

# Age- and sex-related differences in upper-body joint and endpoint kinematics during a drinking task in healthy adults

Jun Nakatake[1], Hideki Arakawa[1], Takuya Tajima[2], Shigeaki Miyazaki[1] and Etsuo Chosa[2]

[1] Rehabilitation Unit, University of Miyazaki Hospital, Miyazaki, Miyazaki, Japan
[2] Department of Orthopaedic Surgery, Faculty of Medicine, University of Miyazaki, Miyazaki, Miyazaki, Japan

## ABSTRACT

**Background:** The objective kinematic assessments of activities of daily living are desired. However, the limited knowledge regarding age- and sex-related differences prevents the adaptation of these measurements to clinical settings and in-home exercises. Therefore, this study aimed to determine the effects of age and sex on joint and endpoint kinematics during a common activity of daily living, specifically, drinking from a glass.

**Methods:** In total, 32 healthy adults (18 males and 14 females) aged 22–77 years performed a drinking task comprising reaching for a glass, bringing it forward and sipping, returning it, and placing the hand back to the starting position, which was recorded using a three-dimensional motion-capturing system. A two-way analysis of variance was used to statistically compare joint angles at five different time points and endpoint kinematic variables in the four drinking phases between older and younger age groups and sexes.

**Results:** Wrist radial deviation was greater in older adults than in younger participants at all five different time points ($F = 5.16$–$7.34$, $p \leq 0.03$, $\eta^2 = 0.14$–$0.21$). Moreover, lesser shoulder abduction and greater shoulder internal rotation and forearm pronation when moving and returning the hand to the starting position were observed in the female group than in the male group ($F = 4.21$–$20.03$, $p \leq 0.0497$, $\eta^2 = 0.13$–$0.41$). Trunk flexion was lower in the female group than in the male group at all time points ($F = 4.25$–$7.13$, $p \leq 0.0485$, $\eta^2 = 0.12$–$0.19$). Regarding endpoint kinematics, the performance time in the reaching phase was longer in older adults than in younger individuals ($F = 4.96$, $p = 0.03$, $\eta^2 = 0.14$). Furthermore, a shorter time while returning the hand to the starting position was observed in the female group than in the male group ($F = 9.55$, $p < 0.01$, $\eta^2 = 0.22$).

**Conclusions:** The joint kinematics of drinking were partially characterized by an age effect, whereas endpoint kinematics were scattered in all drinking phases. Sex-related effects in most upper-body motions and postures may cause rapid motions in females. Therefore, clinicians could use this knowledge for precise assessments and to suggest feasible in-home exercises.

Corresponding author
Jun Nakatake,
jyun_nakatake@med.miyazaki-u.ac.jp

## INTRODUCTION

Three-dimensional motion analysis enables an objective and precise assessment of activities of daily living (ADL). This assessment provides insight into the kinematic features of able and disabled bodies, frequently measured using optical marker-based motion-capturing systems and presented as joint angle and endpoint trajectory data (*Mesquita et al., 2019a*). Moreover, this method is considered a gold standard and facilitates the evaluation of daily movements in patients.

Although various ADL tasks have been analyzed, such as feeding and handling objects (*Mesquita et al., 2019b*), the drinking movements in healthy individuals (*Alt Murphy et al., 2006*; *Mesquita et al., 2020*) and patients with spinal cord injury (*de los Reyes-Guzmán et al., 2010*, *2017*; *Lili et al., 2021*) and cerebral vascular accidents (*Alt Murphy, Willén & Sunnerhagen, 2011*, *2012*, *2013*; *Thrane, Alt Murphy & Sunnerhagen, 2018*, *2020*; *Kim et al., 2014*; *Santos et al., 2018*) have been well-established regarding the involved movement processes and strategies. Additionally, drinking tasks can be performed under realistic conditions without simulation because they do not require special equipment (*Mesquita et al., 2019b*). Therefore, performing this purposeful movement is feasible in an experimental setting for analyzing the motion of healthy and diseased individuals. Furthermore, motion analysis of the drinking movement can distinguish clinically significant improvements in movement capacity (*Alt Murphy, Willén & Sunnerhagen, 2013*). The drinking movement is associated with self-receiver upper limb movement capacity in patients with stroke (*Thrane, Alt Murphy & Sunnerhagen, 2018*). Moreover, drinking movement is believed to largely overlap with eating, brushing teeth, and grooming because they also require reaching toward the face.

Motion capture using applications under continuous development that employ simple systems allows for effective assessment and feedback on patient drinking movements during in-home exercises (*Lee et al., 2019*; *Cóias, Lee & Bernardino, 2022*). The drinking movement has been adopted as a reference exercise in previous studies, where patients were assessed on their motions, and relevant feedback was obtained for improvement, resulting in mixed patient satisfaction. Although the assessment based on the effects of age and sex may be necessary, evidence for age- and sex-related differences in joint and endpoint kinematics during a drinking task remains insufficient.

Clarifying the effects of age and sex on this movement, which indicates patients' clinical status, will improve the feasibility of related kinematic assessments in clinical settings. Moreover, in assessing individuals with diseases, judgments of what is considered normal or otherwise are derived based on comparisons with normal values. Therefore, if age- or sex-related differences exist in the kinematic movements involved in drinking from a glass, such differences should be considered for improved healthcare treatment.

*Lee et al. (2007)* reported lower shoulder joint flexion angles and a more anteriorly tilted pelvic position in older adults when reaching for a table in front of them and their mouth and head during simulated daily tasks than in younger adults. Sex-related differences in ADL kinematics are also expected because of the heterogenous sex ratio between younger and older groups. Studies on eating movements have indicated larger upper limb and

smaller neck motions in young females than in young males (*Inada et al., 2012*; *Nakatake et al., 2017*). Furthermore, sex-related differences in joint kinematics have also been examined only during reaching for a drink on a table in the drinking task, showing increased elbow flexion, decreased wrist extension, and no changes in shoulder joint angles in younger and older females, compared with those in males of all ages (*Mesquita et al., 2020*). Another study revealed that older females demonstrated slowness and a shorter time-to-peak velocity in the drinking movement than did younger females (*Maitra & Junkins, 2004*).

Therefore, we hypothesized sex-related differences in joint kinematics and age-related differences in endpoint kinematics during all stages of the drinking task. In this study, we proposed the importance of age- and sex-related differences in drinking task movements across all related phases for assisting clinicians and engineers in designing appropriate intervention/rehabilitation strategies. This study could provide knowledge about normal kinematics during meaningful ADL in each age and sex group for objective and standardized understanding by researchers and clinicians, as well as an opportunity for evaluating these factors in other daily tasks. Furthermore, patients could be assessed using motion capture systems involving suitable and simplified devices for practical use. Therefore, this study aimed to clarify age- and sex-related differences in upper-body kinematics at joint and endpoint levels during the ADL task of drinking from a glass in healthy adults.

## MATERIALS AND METHODS

### Participants

The participants were 32 healthy community-dwelling individuals (age, 48.3 ± 17.6 years; age range, 22–77 years). The younger and older age groups comprised 16 individuals each ($\chi^2 = 0.00$, $p = 1.00$, $\varphi = 0.00$). The male and female groups had 18 and 14 participants, respectively ($\chi^2 = 0.00$, $p = 1.00$, $\varphi = 0.00$). No sample size calculation was conducted. The power to detect the main effects or interactions was 0.44–1.00 and that to detect the effects of the combined age and sex factors was 0.68–0.99. The Research Ethics Committee of the Faculty of Medicine at the University of Miyazaki (Miyazaki-shi, Japan) approved the study protocol (approval number: O-0459).

### Sample selection criteria

The inclusion criteria were age ≥20 years, absence of nerve and musculoskeletal disorders, self-reporting as "healthy," ability to understand the study's plan and instructions, participating voluntarily, and providing written consent. Current athletes were excluded from this study.

### Instruments

Upper-body movements during drinking were captured using the Vicon Nexus system (Oxford Metrics Ltd., Oxford, UK) and 12 infrared cameras with a sampling rate of 100 Hz. The kinematic model Plug-in-Gait-Fullbody (*Vicon Motion Systems Limited, 2022*) was used to assess the shoulder, wrist, neck, and trunk joints with three degrees of

freedom (DoFs) and the elbow joint with 1 DoF. The model was validated using goniometric measurements (correlation coefficient ≥0.76) (*Henmi et al., 2006*) and based on intra- and inter-rater assessments (coefficients of multiple correlation ≥0.95) (*Chin, 2009*). Thirty-five reflective infrared markers were attached to the bilateral upper and lower limbs, head, and trunk. Position data were processed using a Butterworth filter with a cut-off of 6 Hz. The drinking task was performed with a translucent hard plastic glass with a top diameter, bottom diameter, and height of 8.4, 5.3, and 11.2 cm, respectively.

## Drinking task

The task was to drink a sip of water from a glass using the dominant hand, which was standardized in this study based on a previously reported protocol (*Santos et al., 2018*). The starting and ending positions were a comfortable upright sitting position on a stool without a backrest (the height of the stool was adapted to the participants' lower leg length) and both hands on the lap. A table was placed in front of the knees at elbow height. Next, the participants were asked to drink approximately 5 mL of water four times at a self-selected speed. The first attempt was a mock trial intended to familiarize the participant with the motion, whereas the subsequent three attempts were used for data analysis.

## Hand dominance, body mass index measurements, and grip strength

Hand dominance for daily drinking movement was confirmed verbally. Body mass index (BMI) was calculated using an automatic height and weight scale (DC-250, Tanita Corporation, Tokyo, Japan). Dominant handgrip strength was measured once while standing using a digital dynamometer (TKK-5401, Takei Scientific Instruments Co., Ltd., Niigata, Japan). No related cut-off values were set in this study.

## Kinematic data processing

The drinking task was categorized into four phases as follows (Fig. 1): from the starting position to reaching and grasping the glass (reaching), bringing it to the mouth and drinking (bringing forward), returning the glass to the table (returning), and releasing the glass and returning to the starting position (withdrawing) (*Santos et al., 2018*). Kinematic data were derived by confirming the motions of the stick figure and the reflective marker of the dominant hand.

The joint angles of the kinematic model analyzed were at flexion/extension, abduction/adduction, and internal/external rotation at the shoulder; flexion/extension at the elbow; pronation/supination at the forearm; dorsal/palmer flexion and radial/ulnar deviation at the wrist; and flexion/extension (+/- for all) at the neck and trunk. Specifically, the joint angles at the starting point (SPT) in the reaching phase and endpoint (achieving point; APT) in each phase (Fig. 1) were used for analyzing joint kinematics to survey the participants' movements and postures.

The endpoint kinematics analyzed for the dominant hand are presented as performance times for all drinking phases. In each phase, the following were calculated: ratio of the performance time of each phase to the total performance time for movement, maximum

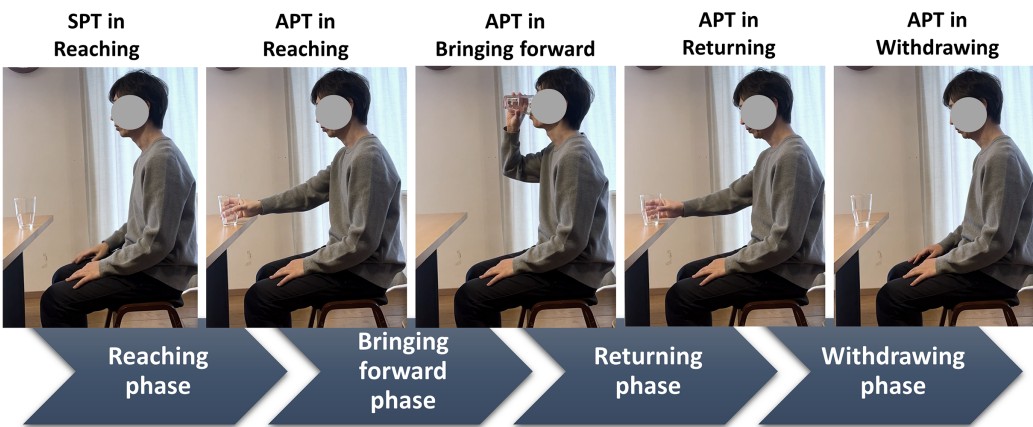

| SPT in Reaching | APT in Reaching | APT in Bringing forward | APT in Returning | APT in Withdrawing |

Reaching phase → Bringing forward phase → Returning phase → Withdrawing phase

**Figure 1 Defined drinking phases and related time points.** The drinking phases are classified by time events (*i.e.*, time points). A drinking movement cycle is considered to begin with the SPT, corresponding to the reaching phase. The cycle is subsequently classified into APT phases (*i.e.*, reaching, bringing forward, returning, and withdrawing). The SPT indicates the starting time point, and the APT indicates the endpoint (*i.e.*, achieving time point). SPT, starting point; APT, achieving point.

velocity, and time to reach maximum velocity; trajectory straightness for smoothness calculated by dividing the actual trajectory distance by the displacement between the positions at SPT and APT; and normalized integrated jerk (NIJ) for smoothness. The NIJ is the change in acceleration normalized by duration and distance (*Santos et al., 2018*; *Deng et al., 2021*).

The joint and endpoint kinematics of the last three repetitions were averaged for each individual. However, the last withdrawing phase was not recorded for one young man; therefore, his data in this phase were recorded as the mean of the second and third cycles.

## Statistical analysis

The dependent variables, which were the grip strength, BMI, and joint and endpoint kinematics (mean ± standard deviation), were compared using a two-way analysis of variance (ANOVA) against the independent variables, age and sex. Age groups were determined based on the number of participants whose age was less or greater than the mean age of all participants, resulting in an equal division of 16 participants, each in the younger and older groups. The participants were also categorized into the male and female groups. *Post-hoc* analysis of multiple comparisons was performed using Tukey's honestly significant difference test. All statistical analyses were conducted using JMP 16 (SAS Institute Inc., Cary, NC, USA). Power analysis of the two-way ANOVA and *post-hoc* test was conducted using G Power 3.1.9.7 (University of Duesseldorf, Duesseldorf, Germany). Statistical significance was set at $p < 0.05$. Furthermore, the effect sizes of the kinematic variables were presented as $\eta^2$ for main effects and interactions and Cohen's $d$ for *post-hoc* comparisons between the groups. A large effect was defined as $\eta^2 = 0.14$ and $d = 0.80$ (*Fritz, Morris & Richler, 2012*).
## RESULTS

### Group characteristics

Age had a significant main effect in the comparison between the younger (age, $33.0 \pm 7.5$ years) and older groups (age, $63.7 \pm 8.8$ years) ($F = 102.80$, $p < 0.01$, $\eta^2 = 0.77$). However, the main effect ($F = 0.05$, $p = 0.83$, $\eta^2 = 0.00$) and interaction ($F = 0.29$, $p = 0.59$, $\eta^2 = 0.00$) were not significant for sex. Only one older male participant was left-handed, whereas the other participants were right-handed. BMI had no significant main effect (age, $F < 0.01$, $p = 1.00$, $\eta^2 = 0.00$; sex, $F = 2.85$, $p = 0.10$, $\eta^2 = 0.09$) or interaction ($F = 0.18$, $p = 0.68$, $\eta^2 = 0.01$). Grip strength had a significant main effect when the male ($42.9 \pm 7.1$ kg) and female ($24.6 \pm 3.7$ kg) groups were compared ($F = 81.43$, $p < 0.01$, $\eta^2 = 0.72$), but no significant effect ($F = 3.26$, $p = 0.08$, $\eta^2 = 0.03$) or interaction ($F = 0.13$, $p = 0.72$, $\eta^2 = 0.00$) for age. Table 1 presents the main characteristics of all groups.

### Joint kinematics

#### *Age-related effects*

Age had significant main effects and large effect sizes for wrist radial deviation at all time points (Fig. 2 and Tables S1 and S2). The older group had greater angles than the younger group at the SPT in the reaching phase ($41.1° \pm 8.2°$ *vs.* $33.9° \pm 8.9°$, $F = 5.30$, $p = 0.03$, $\eta^2 = 0.16$) and APT in the reaching ($38.3° \pm 10.2°$ *vs.* $30.7° \pm 7.2°$, $F = 6.45$, $p = 0.02$, $\eta^2 = 0.17$), bringing forward ($50.6° \pm 12.3°$ *vs.* $40.3° \pm 8.3°$, $F = 7.34$, $p = 0.01$, $\eta^2 = 0.21$), returning ($36.3° \pm 10.6°$ *vs.* $28.0° \pm 9.6°$, $F = 5.16$, $p = 0.03$, $\eta^2 = 0.14$), and withdrawing ($40.3° \pm 8.6°$ *vs.* $31.7° \pm 10.2°$, $F = 5.95$, $p = 0.02$, $\eta^2 = 0.17$) phases.

#### *Sex-related effects*

Several upper-body joint angles had significant main effects and large effect sizes for sex (Fig. 3 and Tables S1 and S2). For the shoulder joint, the abduction angles were lower in the female group than in the male group at the SPT in the reaching phase ($8.8° \pm 4.3°$ *vs.* $13.6° \pm 4.8°$, $F = 8.29$, $p = 0.01$, $\eta^2 = 0.22$) and APT in the reaching ($4.9° \pm 6.3°$ *vs.* $13.3° \pm 5.0°$, $F = 16.81$, $p < 0.01$, $\eta^2 = 0.37$), returning ($7.3° \pm 5.7°$ *vs.* $17.9° \pm 5.0°$, $F = 29.38$, $p < 0.01$, $\eta^2 = 0.51$), and withdrawing ($8.9° \pm 4.5°$ *vs.* $13.9° \pm 4.8°$, $F = 8.81$, $p = 0.01$, $\eta^2 = 0.23$) phases. Moreover, the internal rotation angles were larger in the female group than in the male group at the SPT in the reaching phase ($39.5° \pm 7.3°$ *vs.* $26.7° \pm 8.5°$, $F = 19.09$, $p < 0.01$, $\eta^2 = 0.40$) and APT in the bringing forward ($-7.6° \pm 28.0°$ *vs.* $-35.9° \pm 18.6°$, $F = 12.67$, $p < 0.01$, $\eta^2 = 0.28$) and withdrawing ($40.0° \pm 7.5°$ *vs.* $26.6° \pm 8.6°$, $F = 20.03$, $p < 0.01$, $\eta^2 = 0.41$) phases. For the elbow flexion, the female group had a lower angle than the male group at the APT in the reaching ($70.5° \pm 7.2°$ *vs.* $78.1° \pm 8.0°$, $F = 7.48$, $p = 0.01$, $\eta^2 = 0.20$) and returning ($68.7° \pm 5.7°$ *vs.* $76.8° \pm 8.3°$, $F = 9.53$, $p < 0.01$, $\eta^2 = 0.24$) phases.

Forearm pronation angles were greater in the female group than in the male group at the SPT in the reaching phase ($146.9° \pm 10.6°$ *vs.* $138.6° \pm 11.4°$, $F = 4.21$, $p = 0.0497$, $\eta^2 = 0.13$) and APT in the withdrawing phase ($147.8° \pm 10.5°$ *vs.* $138.4° \pm 12.9°$, $F = 4.63$, $p = 0.04$, $\eta^2 = 0.14$). At the wrist joint, only radial deviation at the APT during the reaching

**Table 1 Participant characteristics in the age and sex groups and in the interaction group.**

| Variable | Age group | | Sex group | | Age + sex group | | | |
|---|---|---|---|---|---|---|---|---|
| | Younger | Older | Male | Female | Younger male | Younger female | Older male | Older female |
| N | 16 | 16 | 18 | 14 | 9 | 7 | 9 | 7 |
| Age range (y) | 22–46 | 51–77 | 22–77 | 23–72 | 22–41 | 23–46 | 51–77 | 51–72 |
| Age (y) | **33.0 ± 7.5**[a] | **63.7 ± 8.8**[a] | 48.1 ± 18.6 | 48.7 ± 16.8 | 32.0 ± 7.3 | 34.3 ± 8.2 | 64.1 ± 10.1 | 63.1 ± 7.7 |
| BMI (kg/m$^2$) | 22.8 ± 3.1 | 22.7 ± 1.9 | 23.4 ± 2.6 | 21.9 ± 2.2 | 23.6 ± 3.5 | 21.7 ± 2.1 | 23.2 ± 1.4 | 22.1 ± 2.3 |
| Grip strength (kg) | 36.8 ± 11.1 | 33.0 ± 10.7 | **42.9 ± 7.1**[b] | **24.6 ± 3.7**[b] | 45.1 ± 6.6 | 26.0 ± 3.4 | 40.7 ± 7.2 | 23.1 ± 3.6 |

Notes:
Data are presented as the mean ± SD. Bold text indicates $p < 0.05$.
[a] Significant main effect for age ($p < 0.05$).
[b] Significant main effect for sex ($p < 0.05$).
SD, standard deviation; BMI, body mass index.

phase was greater in the female group than in the male group (38.0° ± 9.0° *vs.* 31.8° ± 9.2°, $F = 4.28$, $p = 0.0479$, $\eta^2 = 0.11$).

Trunk flexion angles were lower in the female group than in the male group at all time points: SPT in the reaching phase (17.1° ± 9.9° *vs.* 24.2° ± 8.0°, $F = 5.14$, $p = 0.03$, $\eta^2 = 0.14$) and APT in the reaching (18.4° ± 10.2° *vs.* 24.9° ± 7.9°, $F = 4.25$, $p = 0.0485$, $\eta^2 = 0.12$), bringing forward (15.3° ± 9.6° *vs.* 23.3° ± 7.7°, $F = 7.13$, $p = 0.01$, $\eta^2 = 0.19$), returning (18.2° ± 10.3° *vs.* 25.3° ± 7.7°, $F = 5.02$, $p = 0.03$, $\eta^2 = 0.14$), and withdrawing (17.6° ± 9.7° *vs.* 24.7° ± 8.0°, $F = 5.40$, $p = 0.03$, $\eta^2 = 0.15$) phases.

### Effects of combining age and sex

Some significant interactions occurred at the APT in the bringing forward phase, as shown in Figs. 2 and 3 and Tables S1 and S2. Shoulder flexion had a significant interaction ($F = 7.10$, $p = 0.01$, $\eta^2 = 0.16$). *Post-hoc* analysis (Fig. 4 and Tables S1 and S2) indicated a smaller angle in older males (59.7° ± 5.5°) than in younger males (66.2° ± 3.8°; $t = 2.84$, $p = 0.04$, $d = 1.37$) and older females (68.8° ± 3.0°; $t = 3.74$, $p < 0.01$, $d = 2.00$). Shoulder abduction had a significant interaction ($F = 5.41$, $p = 0.03$, $\eta^2 = 0.10$). The shoulder abduction angle was smaller in younger females (28.4° ± 27.8°) than in younger (81.2° ± 16.9°; $t = -4.80$, $p < 0.01$, $d = 2.38$) and older (66.8° ± 16.6°; $t = -3.49$, $p = 0.01$, $d = 1.74$) males. Additionally, the shoulder abduction angle was smaller in older females (50.2° ± 26.7°) than in younger males (81.2° ± 16.9°; $t = -2.82$, $p = 0.04$, $d = 1.43$). Furthermore, the neck flexion/extension angle showed a significant interaction ($F = 10.39$, $p < 0.01$, $\eta^2 = 0.24$). *Post-hoc* analysis showed that younger males (60.2° ± 6.6°) extended their neck joints more than younger females (44.6° ± 11.4°; $t = 3.12$, $p = 0.02$, $d = 1.74$) and older males did (43.5° ± 9.6°; $t = -3.58$, $p = 0.01$, $d = 2.02$).

## Endpoint kinematics

### Age-related effects

The total time had no significant main effect for age ($F = 3.74$, $p = 0.06$, $\eta^2 = 0.11$). The older group required more time than the younger group during the reaching ($F = 4.96$, $p = 0.03$, $\eta^2 = 0.14$) and withdrawing ($F = 4.99$, $p = 0.03$, $\eta^2 = 0.12$) phases, whereas the time ratio differences between the age groups in all phases were not significant (all phases,

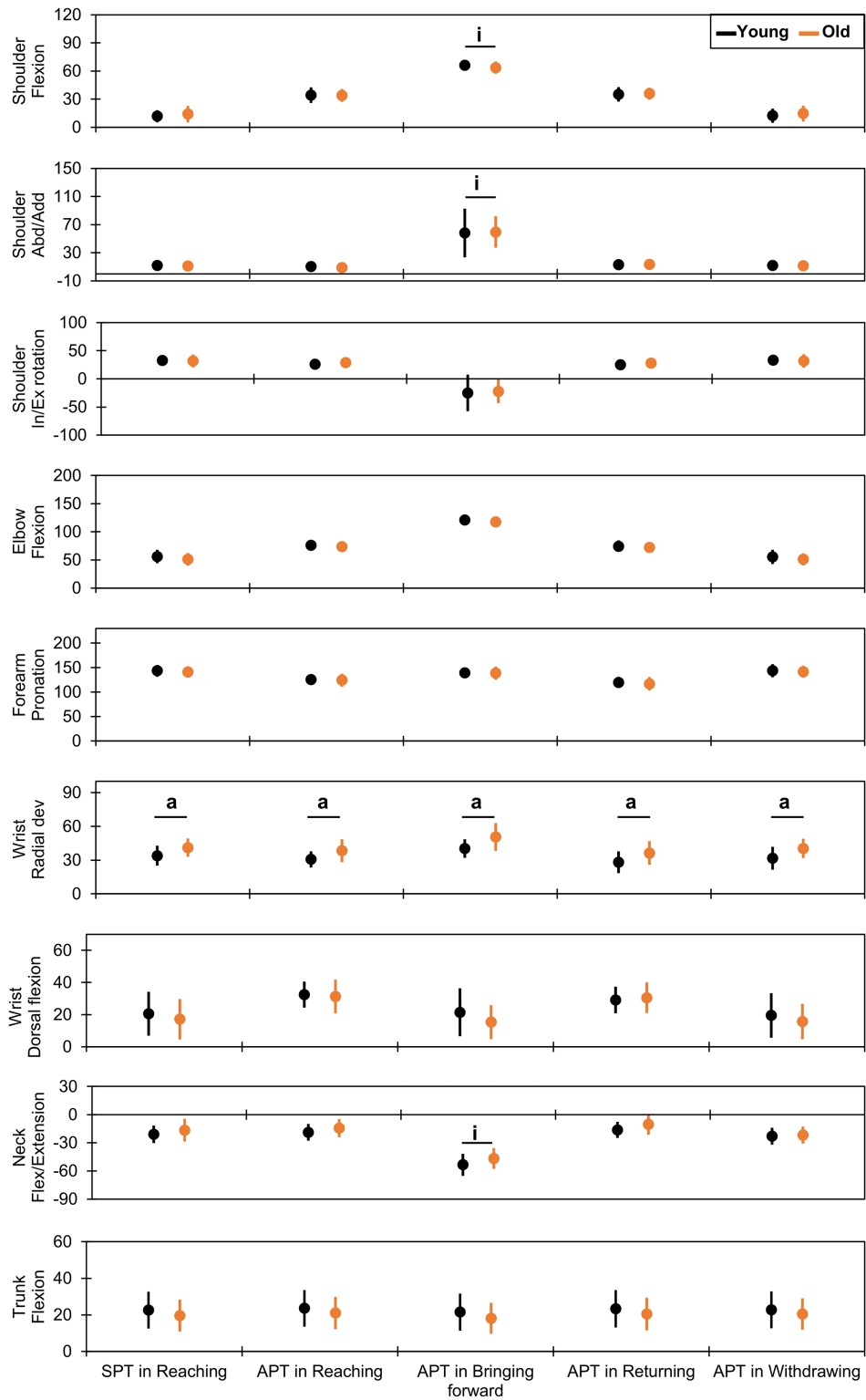

**Figure 2 Comparison of joint kinematics between the younger and older groups.** The vertical and horizontal axes indicate the joint angles in degrees and the time points for each drinking phase, respectively. The dots and error bars indicate the mean and standard deviation, respectively. The underlined "a" and "i" indicate a statistically significant main effect for age and significant interaction between age and sex, respectively. SPT indicates the starting time point, and APT indicates the endpoints (achieving time points). Abd/Add, abduction/adduction; In/Ex, internal/external; dev, deviation; Flex, flexion; SPT, starting point; APT, achieving point.

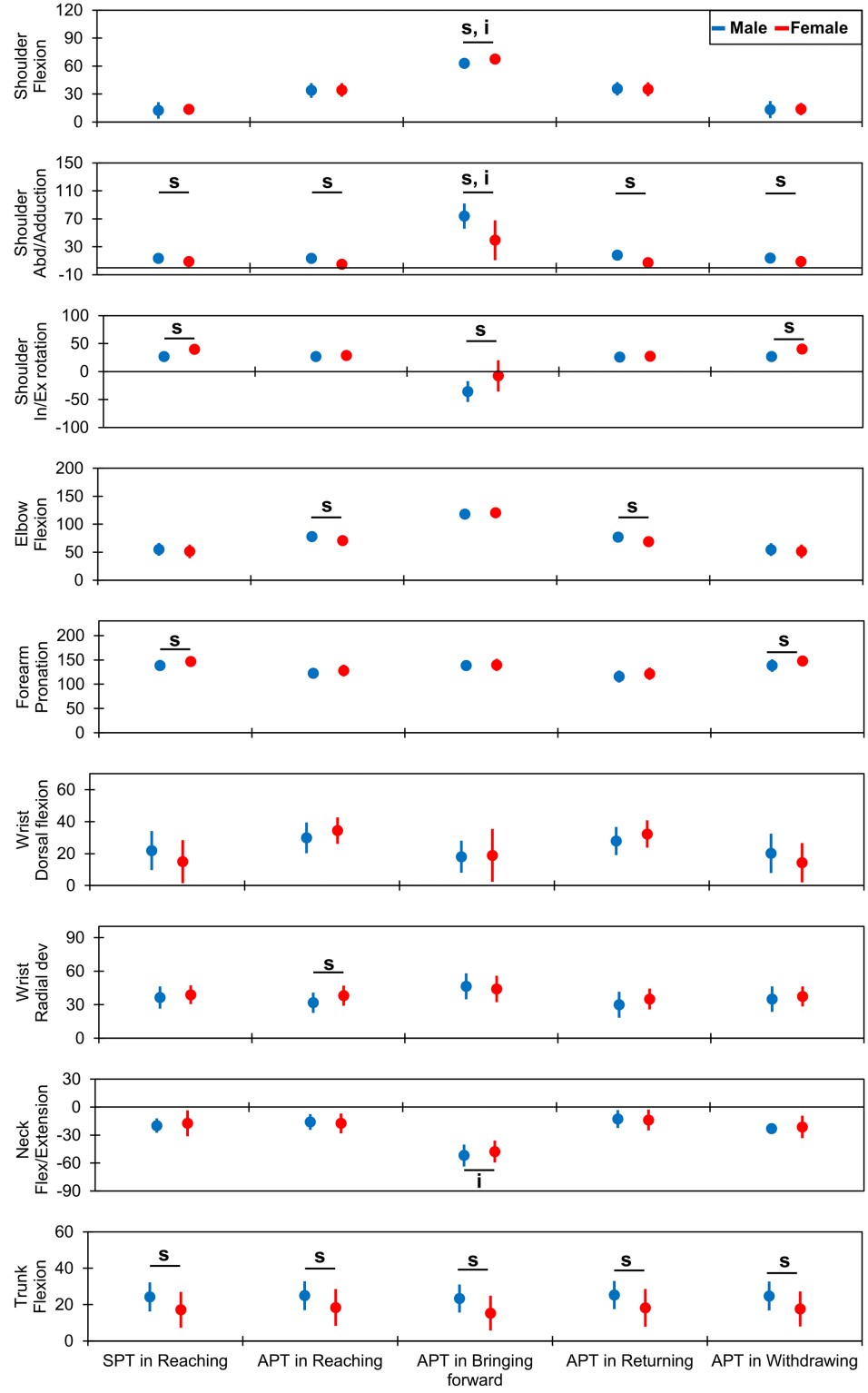

**Figure 3 Comparison of joint kinematics between the male and female groups.** The vertical and horizontal axes indicate the joint angles in degrees and the time points for each drinking phase, respectively. The dots and error bars indicate the mean and standard deviation, respectively. The underlined "s" and "i" indicate a statistical significance in the main effect of sex and significance in the interaction between age and sex, respectively. SPT indicates the starting time point, and APTs indicate the endpoints (achieving time points). Abd, abduction; In/Ex, internal/external; Flex, flexion; dev, deviation; SPT, starting point; APT, achieving point.

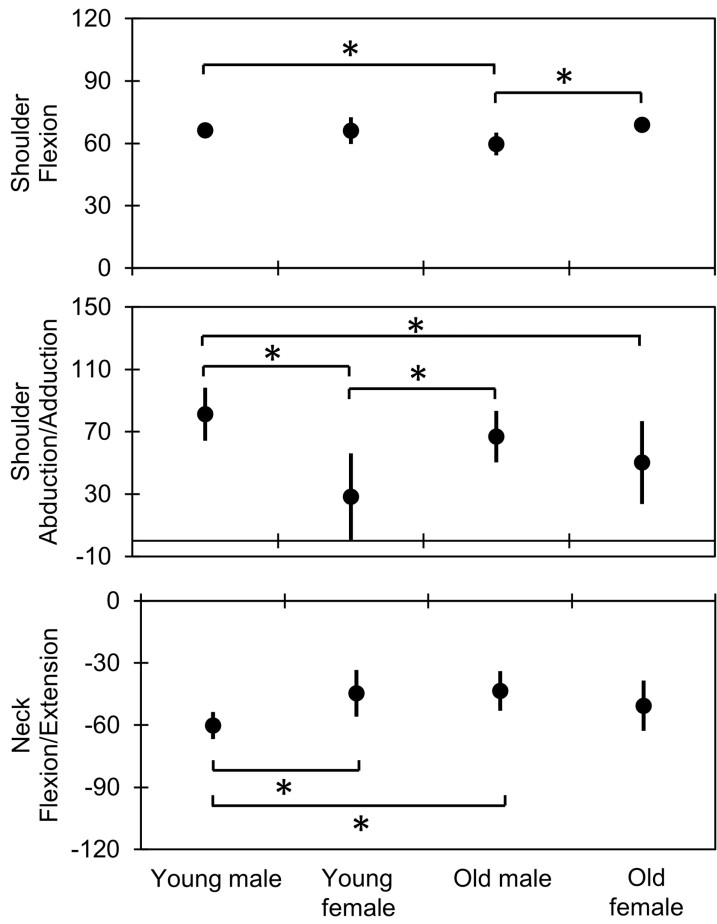

**Figure 4 Comparison of joint kinematics at the APTs in the bringing forward phase between the combined age and sex groups.** The vertical and horizontal axes indicate the joint angles in degrees and the groups, respectively. The dots and error bars indicate the mean and the standard deviation, respectively. An asterisk (*) indicates statistical significance, based on the *post-hoc* honestly significant difference test ($p < 0.05$). APT, achieving point.

$F \leq 2.51$, $p \geq 0.12$, $\eta^2 \leq 0.08$). Peak velocity in the bringing forward phase was lower in the older group than in the younger group ($F = 5.93$, $p = 0.02$, $\eta^2 = 0.17$). The time-to-peak velocity in the older group during the reaching phase was faster than that in the younger group ($F = 4.81$, $p = 0.04$, $\eta^2 = 0.13$). Furthermore, the trajectory straightness value for smoothness of the older group during the returning phase was significantly higher than that of the younger group ($F = 5.07$, $p = 0.03$, $\eta^2 = 0.15$). Moreover, the NIJ value during the reaching phase was higher in the older group than in the younger group ($F = 6.42$, $p = 0.02$, $\eta^2 = 0.17$) (Tables 2, S3 and S4).

### Sex-related effects

The total time had no significant main effect for sex ($F = 0.71$, $p = 0.41$, $\eta^2 = 0.02$). The female group had a shorter performance time than the male group during the withdrawing phase ($F = 9.55$, $p < 0.01$, $\eta^2 = 0.22$). Therefore, the time ratio of the female group was higher during the bringing forward phase ($F = 9.43$, $p < 0.01$, $\eta^2 = 0.24$) and lower during the withdrawing phase ($F = 8.27$, $p = 0.01$, $\eta^2 = 0.22$) than that of the male

**Table 2 Comparison of endpoint kinematics in the groups, classified by age and sex.**

| Variables | Phases | Age group | | Sex group | |
|---|---|---|---|---|---|
| | | Younger | Older | Male | Female |
| **Total time (seconds)** | Total | 6.00 ± 1.06 | 6.78 ± 1.07 | 6.53 ± 1.17 | 6.21 ± 1.07 |
| **Time (seconds)** | Reaching | **1.01 ± 0.22**[a] | **1.19 ± 0.21**[a] | 1.15 ± 0.24 | 1.03 ± 0.21 |
| | Bringing forward | 2.17 ± 0.61 | 2.41 ± 0.52 | 2.23 ± 0.57 | 2.37 ± 0.57 |
| | Returning | 1.67 ± 0.31 | 1.80 ± 0.26 | 1.76 ± 0.29 | 1.71 ± 0.30 |
| | Withdrawing | **1.16 ± 0.26**[a] | **1.39 ± 0.37**[a] | **1.41 ± 0.36**[b] | **1.10 ± 0.20**[b] |
| **Time ratio (%)** | Reaching | 17.0 ± 2.9 | 17.5 ± 2.1 | 17.7 ± 2.7 | 16.7 ± 2.2 |
| | Bringing forward | 35.9 ± 4.8 | 35.4 ± 3.6 | **33.9 ± 3.8**[b] | **37.9 ± 3.6**[b] |
| | Returning | 28.1 ± 2.8 | 26.6 ± 2.4 | 27.2 ± 3.1 | 27.6 ± 2.1 |
| | Withdrawing | 18.9 ± 3.6 | 20.5 ± 3.7 | **21.2 ± 3.8**[b] | **17.8 ± 2.3**[b] |
| **Peak velocity (mm/second)** | Reaching | 858 ± 139 | 824 ± 120 | 864 ± 144 | 811 ± 105 |
| | Bringing forward | **728 ± 110**[a] | **622 ± 124**[a] | 667 ± 125 | 684 ± 134 |
| | Returning | 849 ± 145 | 817 ± 140 | 799 ± 147 | 829 ± 110 |
| | Withdrawing | 832 ± 174 | 776 ± 108 | 838 ± 98 | 807 ± 212 |
| **Time-to-peak velocity (%)** | Reaching | **32.2 ± 8.6**[a] | **26.3 ± 5.5**[a] | 30.8 ± 8.7 | 27.3 ± 5.9 |
| | Bringing forward | 18.3 ± 2.6 | 18.3 ± 3.6 | 18.9 ± 3.3 | 17.5 ± 2.7 |
| | Returning | 45.7 ± 5.1 | 42.2 ± 6.7 | 42.4 ± 6.6 | 46.0 ± 5.1 |
| | Withdrawing | 55.6 ± 6.8 | 56.9 ± 9.5 | 55.1 ± 7.5 | 57.8 ± 9.0 |
| **Trajectory straightness** | Reaching | 1.319 ± 0.116 | 1.464 ± 0.279 | 1.430 ± 0.269 | 1.343 ± 0.137 |
| | Bringing forward | **1.019 ± 0.011**[c] | **1.022 ± 0.007**[c] | **1.017 ± 0.008**[b,c] | **1.026 ± 0.008**[b,c] |
| | Returning | **1.039 ± 0.020**[a] | **1.055 ± 0.020**[a] | 1.045 ± 0.020 | 1.050 ± 0.023 |
| | Withdrawing | 1.443 ± 0.377 | 1.507 ± 0.151 | 1.562 ± 0.345 | 1.364 ± 0.116 |
| **NIJ** | Reaching | **130,140 ± 66,402**[a] | **202,674 ± 90,792**[a] | 186,479 ± 91,248 | 140,600 ± 75,322 |
| | Bringing forward | 572,448 ± 473,116 | 637,327 ± 373,511 | 562,235 ± 415,410 | 659,725 ± 436,371 |
| | Returning | 302,286 ± 129,287 | 356,147 ± 123,532 | 337,182 ± 137,924 | 318,974 ± 116,631 |
| | Withdrawing | 204,760 ± 144,677 | 344,424 ± 316,306 | **361,608 ± 308,046**[b] | **162,715 ± 59,533**[b] |

Notes:
  Data are presented as mean ± SD. Bold text indicates $p < 0.05$.
  [a] Significant main effect for the age factor.
  [b] Significant main effect for the sex factor.
  [c] Significant interaction effect for age and sex factors.
  SD, standard deviation; NIJ, normalized integrated jerk.

group. Differences in peak velocity and time-to-peak velocity between sexes were not significant during any phase ($F \leq 2.99$, $p \geq 0.09$, $\eta^2 \leq 0.09$). The NIJ value during the withdrawing phase was smaller in the female group than in the male group ($F = 5.92$, $p = 0.02$, $\eta^2 = 0.16$).

### Combining the effects of age and sex

Endpoint kinematics had an interaction only in the trajectory straightness value during the bringing forward phase ($F = 6.82$, $p = 0.01$, $\eta^2 = 0.14$). The values in younger (1.03 ± 0.01) and older (1.02 ± 0.004) females were higher than those in younger males (1.01 ± 0.01; for younger males *vs.* younger females: $t = 4.23$, $p < 0.01$, $d = 1.95$; for younger males *vs.* older females: $t = 2.93$, $p = 0.03$, $d = 1.31$).

## DISCUSSION

This study aimed to determine the effects of age and sex on joint and endpoint kinematics during a purposeful ADL task of drinking from a glass. The kinematic analysis of all phases of drinking demonstrated sex-related differences in various joint angles and age- and sex-related variations in some endpoint variables with large effects. Additionally, the joint kinematics of the wrist and neck were affected by age and the combination of age and sex, respectively. Our hypothesis that there are sex-related differences in joint kinematics and age-related differences in endpoint kinematics was confirmed. Moreover, some age-related joint and sex-related endpoint kinematic patterns were identified.

Our findings suggest that the kinematic analysis of drinking, which is well-established by researchers and rehabilitation practitioners, should consider age and sex. Furthermore, developing a device for at-home exercises for patients adopting a feasible system is also needed (*Lee et al., 2019*; *Cóias, Lee & Bernardino, 2022*) based on the normal kinematics of each age and sex group.

### Joint kinematics

No age-related effects were detected on the shoulder, elbow, forearm, and trunk kinematics at all time points during all phases. However, when the distal joint angles of the upper limb were analyzed, the wrist joint in older adults deviated more radially than that in younger adults at all time points of drinking. This independent effect of aging is similar to the greater radial deviation in the active range of motion of the wrist, whereas active ranges of motion in many other joints were reduced in older adults compared with those in younger adults (*Hwang & Jung, 2015*). Plausibly, this may be because older and younger adults fix their joint angles laterally and in a neutral position, respectively, although this warrants further investigation. This finding may be attributed to the compensatory strategy of the central nervous system: for decreasing DoFs, older adults may fix their wrist joint in radial deviation (*Seidler, Alberts & Stelmach, 2002*).

The findings regarding the effects of sex on various joint kinematics showed differences at all time points of drinking. First, trunk flexion in females was smaller than that in males across all time points, indicating that the differences in the posture of the proximal body parts, reported as lumbar lordosis in females, are greater than those in males (*Arshad et al., 2019*). Furthermore, at the SPT in the reaching phase and the APT in the withdrawing phase—which occurs in a static sitting posture—smaller angles of abduction, greater internal rotation of the shoulder, and greater pronation of the forearm were noted in females than in males.

Similar to the tendency of the shoulder joint, the angles of abduction at the APT in the reaching and returning phases were lower, and the angle of the internal rotation at the APT in the bringing forward phase was greater in females than in males. These positions near the inner side of the shoulder and forearm in females may be produced by weak upper limb strength, as demonstrated by their grip strength. This factor could also be used to estimate the upper limb (*Bohannon, 2009*) and global (*Porto et al., 2019*) strengths. However, males performed greater motions of the upper extremities unless their mass proportion was larger than that of females (*Whittaker et al., 2021*). This finding suggested no relationship

between the extent of motion and the weight of body parts. Moreover, the present study's results may be interpreted based on an individual's strength.

Additionally, the APTs in the reaching and returning phases were estimated from the forward-reaching movement of the upper limb to compensate for the non-participation of the trunk. Elbow flexion was smaller (*i.e.*, greater extension was observed) in females than in males. Regarding wrist radial deviation at the APT in the reaching phase, females had a greater angle than males, indicating the need for wrist calibration towards an extended position in relation to the elbow. Contrary to our results, a recent study (*Mesquita et al., 2020*) showed that females had greater elbow and wrist flexions than males at the end of the action of reaching for a glass. Nevertheless, coordination was maintained between the elbow and wrist joints, similar to our findings. Joint coordination may be directly proportional to the amount of water drank from a glass, which was particularly scarce in this study compared with that reported in a previous study (*Mesquita et al., 2020*). Therefore, this characteristic should be considered in future studies.

Interestingly, the combined effects of age and sex on joint kinematics were detected only at the APT of the bringing forward phase. This event could be characterized by regulating the mouth and hand positions, represented by the neck and upper limb joint angles. Therefore, when younger males exhibit greater neck extension than younger females and older males, they simultaneously display greater upper arm elevation (*i.e.*, shoulder flexion or abduction). Greater shoulder abduction was similarly evident in younger males than in older females. Although the difference in neck extension was not significant, the mean angles were larger in younger males. The greater extension of the neck (approximately 60°) in younger males is close to the endpoint of the active range of motion for younger adults (*Pan et al., 2018*). Particularly, this finding could also be caused by the greater flexion of the trunk observed in males than in females in this study. Therefore, the greater degree of neck joint extension in younger males may lead to greater upper arm elevation during sipping.

## Endpoint kinematics

Slowness and shorter time-to-peak velocity have been reported in older females (*Maitra & Junkins, 2004*). In this study, the performance times of males and females differed according to age, with the older group needing more time in the reaching and withdrawing phases than the younger group. The peak velocity of the bringing forward phase was also lower in older adults than in younger adults. However, this slowness did not affect the total time or time ratios. Furthermore, the earlier time-to-peak velocity in the reaching phase in the older group indicated an altered strategy with aging. Regarding smoothness (*i.e.*, trajectory straightness and NIJ), individuals in the older group had reduced smoothness during the reaching and returning phases compared with those in the younger group. Consistent with the findings of previous studies (*Cooke, Brown & Cunningham, 1989*; *Ketcham et al., 2002*; *Seidler, Alberts & Stelmach, 2002*), slowness, an altered reaching strategy, and low smoothness in older individuals occurred across all drinking phases. Therefore, the endpoint kinematics of older adults may be partly reflected in their daily activities.

The effect of sex on endpoint kinematics was likely limited to the withdrawing phase, where the movement of females was shorter, with lower time ratio and smaller NIJ values than those of males. These fast and smooth motions in females are consistent with those of the task of switching on the light, as elucidated by *Mesquita et al. (2020)*, who suggested that the lower mass proportion of the upper body in females can cause decreased inertia. In contrast, the findings regarding the combined effects of age and sex on trajectory straightness in the bringing forward phase indicated that younger males moved more smoothly than older females. Therefore, these results should be further explored in the future concerning possible muscle weakness (*Hughes et al., 1999*), although the effects of combining age and sex regarding this phenomenon were not observed in this study.

## Limitations

This study has some limitations. First, the power in this analysis was somewhat low; therefore, the observed effects in the joint and endpoint kinematics might have been partly revealed. Second, age and sex effects on the kinematics of the ADL movement of drinking were elucidated; however, applications of these results to other ADL movements involving tasks requiring larger motion, greater strength, speed, or repetition may be limited. Lastly, the effects of age and sex on drinking kinematics among non-healthy individuals were not investigated. Therefore, future studies should explore these limitations to validate our understanding. Moreover, future studies on drinking kinematics should consider the age and sex factors and develop a feasible system involving a simple device for in-home exercises.

## CONCLUSIONS

The analysis of age- and sex-related effects on upper-body kinematics during a purposeful daily movement of drinking from a glass revealed that older adults radially deviate their wrist constantly and that their slowness, altered reaching strategy, and low smoothness are partly recognized in the endpoint kinematics of the upper limb. Females extend their trunks more, position their upper limbs near the inside, and perform fast and smooth motions. Furthermore, neck extension during sipping is greater in younger males than in younger females and older males. Therefore, the age and sex characteristics presently determined in the well-studied purposeful ADL task of drinking from a glass will provide clinicians with insights into each group's normal postures and movements and facilitate the adaptation of suitable kinematic assessments for rehabilitation. Furthermore, developing a feasible system involving a simple device will promote the adaptation of in-home exercises, making treatment of populations with ADL disorders efficient and cost-effective.

## ACKNOWLEDGEMENTS

The authors thank Shogo Maeda for their assistance with the experimental measurements.

### Funding

This work was supported by the University of Miyazaki Hospital. The funders had no role in study design, data collection and analysis, decision to publish, or preparation of the manuscript.

### Grant Disclosures

The following grant information was disclosed by the authors:
University of Miyazaki Hospital.

### Competing Interests

The authors declare that they have no competing interests.

### Author Contributions

- Jun Nakatake conceived and designed the experiments, performed the experiments, analyzed the data, prepared figures and/or tables, authored or reviewed drafts of the article, and approved the final draft.
- Hideki Arakawa conceived and designed the experiments, authored or reviewed drafts of the article, and approved the final draft.
- Takuya Tajima analyzed the data, prepared figures and/or tables, and approved the final draft.
- Shigeaki Miyazaki analyzed the data, prepared figures and/or tables, and approved the final draft.
- Etsuo Chosa conceived and designed the experiments, authored or reviewed drafts of the article, and approved the final draft.

### Human Ethics

The following information was supplied relating to ethical approvals (*i.e.*, approving body and any reference numbers):

The research ethics committee of the Faculty of Medicine at the University of Miyazaki.

### Data Availability

The raw data is available in the Supplemental Table.

### Supplemental Information

Supplemental information for this article can be found online at http://dx.doi.org/10.7717/peerj.16571#supplemental-information.

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
