# Peer review of "Age- and sex-related differences in upper-body joint and endpoint kinematics during a drinking task in healthy adults"

_PeerJ, doi:10.7717/peerj.16571_

## Round 0.1 · original submission · Major Revisions

The paper is interesting, particularly for researchers and health-related professionals. However, several clarifications are needed. Please ensure that all reviewers' suggestions are addressed, and pay particular attention to i) the need to highlight the relevance and contribution of the study; ii) clarifications on methodology; iii) deeper discussion on limitations; iv) highlighting the practical findings.

**Language Note:** PeerJ staff have identified that the English language needs to be improved. When you prepare your next revision, please either (i) have a colleague who is proficient in English and familiar with the subject matter review your manuscript, or (ii) contact a professional editing service to review your manuscript. PeerJ can provide language editing services - you can contact us at copyediting@peerj.com for pricing (be sure to provide your manuscript number and title). – PeerJ Staff

Reviewer 1 ·

Basic reporting

Dear authors,
first of all thanks for the submission made to PeerJ. The authors sought to determine the effects of age and sex on joint kinematics and the end of a daily activity.
The authors have done an overall good job on a topic that may be of potential importance for designing rehabilitation programs.
Still, there are some points that must be improved before the article can be considered for publication. Below you can find my comments in detail.

The authors provide a sufficiently broad explanation of why they used the drinking task in their study and this seems clear, yet perhaps it would be important to emphasize how the data/results obtained through this type of investigation can be used as a starting point. starting point for the formulation of rehabilitation programs. This is because it is not clear to the reader the potential relationship that the task of drinking water may have with other day-to-day tasks.

Provide study hypotheses.

The bibliographic references used are mostly current and fit the topic of study.

The tables are unattractive in their present form. Remove the lines inside the table and leave only the upper and lower limits (horizontally).

Experimental design

In general, the information provided allows replication of the study.

Please identify how you calculated the required sample and which were the dependent and independent variables of the study.

Add exclusion criteria

Also clarify how the age division of the participants was made

Validity of the findings

I would like to see a specific field added at the end of the article with practical recommendations resulting from the findings of the article. I believe this is a key point to improve the final quality of the manuscript.

Additional comments

No Comment

·

Basic reporting

The aim of this study was to determine the effects of age and sex on joint and endpoint kinematics during a purposeful activity of daily living, i.e., drinking from a glass. It is an interesting manuscript well written and with a robust methodology, however some sections need some improvements.

Experimental design

Please, check the revisions for experimental design and procedures:
• Lines 90–91 (Material & Methods): “[mean ± standard deviation]” – This information should be in the statistical analysis sub-section. Also, the selection criteria should be presented in a separate section (i.e., selection criteria). The variables hand dominance, grip strength and BMI should only be presented after the description of the procedures (sub-section one section for these variables). Also, clarify the materials, procedures and cut-off values used for each variable.
• Lines 101–105 (Material & Methods): Please include the link in the references. At this point you should only describe the ref of the plug-in and the validity/reliability values described in the literature.
• Lines 110–117 (Material & Methods): What guidelines or recommendations did you use to structure the drinking task?
• Lines 151–152 (Material & Methods): Please specify the power of the sample (calculated by G power); you can do this in the participants sub-section.

Validity of the findings

The results are robust and well presented. Please, check the following lines:
• Lines 167–170 (Results): This sentence should be placed in the participants sub-section (i.e.., Material & Methods). Also, add the qualitative effect sizes for Cohen's d and eta squared.

Additional comments

Please, check additional comments to improve Abstract, Introduction, Discussion and Conclusions:
• Line 1–2 (Title): Please, add the research context/target study population.
• Lines 16–18 (Abstract): The authors refer to "objective and exact kinematic assessments". Which one did you use? Please specify.
• Lines 18–19 (Abstract): The kinematic analysis evaluated the activity of daily living "drinking from a glass"? Did you only analyze this activity of daily living?
• Lines 21–25 (Abstract): What methodology did you use for Joint Angles? What test do you use to compare the kinematic variables statistically?
• Lines 26–36 (Abstract): Please provide the range of F statistics (instead of higher/lower or equal). Also, please provide the eta squared (or partial eta squared). Conclusions should be presented separately from the results as sub-topics.
• Lines 39–42 (Introduction): There are several types of three-dimensional motion analysis. You should expand a little more on the type of kinematics analysis that is commonly used, which is the gold standard and why the one used in this study was selected.
• Lines 43–65 (Introduction): What other ADLs have already been analyzed? Why did you choose the drinking movement? Also, the authors refer to various pathologies. This study analyses the age and sex-related differences in upper-body joint and endpoint kinematics during this ADL in which population? Healthy or with a diagnosed pathology. The research gap is not well delimited, please clarify.
• Lines 289–370 (Discussion): The discussion is very well structured and related to the results. However, it should add the research limitations, future perspectives and practical applications.
• Lines 380–390 (Conclusions): The conclusions section should not contain quotations, but should be reserved for the discussion. They should be short and to the point, describing the main outcome (as described in lines 381–386) and the main novelty/practical application for the scientific and practitioner community. What is new about this study and how can we extrapolate these results for the prevention, rehabilitation and treatment of pathologies (or for improving quality of life)?

---

## Round 0.2 · accepted · Accept

The authors have addressed all the reviewers' comments, and I am happy with the current version. In my opinion, this manuscript is ready for publication.

·

Basic reporting

The revised article expresses the effort and rigor of the authors in responding to all the reviews in the first round.

Experimental design

Everything is in order in this revised version of the manuscript. Nothing else to add.

Validity of the findings

Everything is in order in this revised version of the manuscript. Nothing else to add.

Additional comments

Everything is in order in this revised version of the manuscript. Nothing else to add.